# Study of Microwave Heating Effect in the Behaviour of Graphene as Second Phase in Ceramic Composites

**DOI:** 10.3390/ma13051119

**Published:** 2020-03-03

**Authors:** Rut Benavente, María Dolores Salvador, Alba Centeno, Beatriz Alonso, Amaia Zurutuza, Amparo Borrell

**Affiliations:** 1Instituto de Tecnología de Materiales, Universitat Politècnica de València, Camino de Vera, s/n, 46022 Valencia, Spain; dsalva@mcm.upv.es (M.D.S.); aborrell@upv.es (A.B.); 2Graphenea S.A. Paseo Mikeletegi, 83, 20009 San Sebastián, Guipúzcoa, Spain; a.centeno@graphenea.com (A.C.); b.alonso@graphenea.com (B.A.); a.zurutuza@graphenea.com (A.Z.)

**Keywords:** *β*-eucryptite, graphene, microwave sintering, mechanical properties, microstructure

## Abstract

The choice of the right material is essential in microwave processing. The carbon materials are good microwave absorbers, which allows them to be transformed by microwave heating into new carbon materials with adapted properties, capable of heating other materials indirectly. In this paper, the microwave heating of graphene as reinforcement of the lithium aluminosilicate (LAS) ceramics has been explored. LAS ceramics have a near-zero coefficient of thermal expansion and exhibit an effective and efficient heating by microwave. Nevertheless, we have found that the graphene did not show any significant response to the microwave radiation and, hence, the interaction as mechanical reinforcement with the LAS material is harmful. The possible benefits of graphene materials to microwave technology are widely known; however, the mechanism involved in the interaction of microwave radiation with ceramic-graphene composites with high dielectric loss factors has not been addressed earlier.

## 1. Introduction

The lithium aluminosilicate (LAS) system has been extensively studied in the last decades, due to its very low or even negative coefficient of thermal expansion (CTE) [1,2,3,4,5]. Accordingly, LAS materials have found a wide application field including cookware, bakeware, electronic devices, telescope mirror blanks, ring-laser gyroscopes, optically stable platforms and new materials for space missions. Nevertheless, these applications have been focused on the glass-ceramic materials. However, there are very interesting applications in the LAS system that requires obtaining this material in the solid state [4,5,6]. In the last years, to obtain LAS in the solid state with high mechanical properties and a near-zero CTE values has been a great challenge [6,7,8]. The high temperatures required to fully densify LAS powders result in vitreous phase transformation, large grain sizes and consequently decreased mechanical and thermal properties. There are two ways to overcome these problems; to employ non-conventional sintering alternative methods such as spark plasma sintering (SPS) and microwave sintering or to reinforce the LAS matrix with a second phase (i.e., Al_2_O_3_, SiC, carbon nanofibers…) [9,10,11,12].

In previous studies, Benavente et al. demonstrated that the LAS material obtained by microwave sintering, compared with other heating techniques (SPS and conventional), has a lot of benefits in terms of mechanical properties and microstructural design, as well as in terms of processing speed [7]. The important characteristics associated to the microwave process are rapid and uniform volumetric heating, enhancement in densification, improved production rate, and hence it offers significant time savings which can be cost-effective and more importantly clean and environment-friendly [8,13,14,15,16]. Therefore, in this work, on one hand, the microwave heating is proposed as a non-conventional sintering technique to fabricate LAS materials [7,8,13], and on the other hand, a second phase of carbon material, graphene, has been proposed for: (1) studying the microwave effect of graphene as a second phase in the ceramic matrix, since the carbon material is an excellent absorber of microwave radiation [17,18], and (2) investigating graphene as a reinforcement in ceramic materials [19,20].

Other positive effects of graphene as second phase in LAS ceramic matrix could be: to inhibit the grain growth, to decrease the electrical resistivity, to increase the mechanical properties, and to allow the adjustment and the control of the CTE in a wide range of values [12]. Accordingly, these materials could be suitable to be used in new complex components that are required for advanced applications.

In the last decade, the use of carbon nanotubes (CNT) or carbon nanofibers (CNF) to reinforce the ceramic matrix has been extensively studied, as manifested in the high number of publications in this subject, where the mechanical and electrical properties in comparison to the monolithic material were improved [12,21,22]. Graphene has attracted a great attention worldwide due to its unique combination of electrical [23], mechanical [24] and thermal [25] properties. Thus, graphene is an important second phase in order to potentially improve simultaneously the mechanical, electrical, and thermal properties of metals [26], polymers [27], and ceramics [19]. Certainly, the main advantage of using graphene instead of CNTs or CNFs as reinforcement material is its morphology (2D), which is less likely to lead to agglomeration, and has better dispersion in the ceramic matrix [20].

Carbon materials have proven to be excellent absorbers of microwave, being easily heated by microwave radiation. This feature allows them to be used indirectly to heat the matrix in which they are distributed, or to act as a catalyst in various heterogeneous receptor reactions [17,18].

Therefore, the main goal of this work is to design LAS-graphene composites by non-conventional microwave sintering techniques at relatively low temperatures and with short sintering times. The sintering capability by microwave radiation, the mechanical and electrical properties together with the thermal stability of the composite in a wide temperature range (−150 to 450 °C) are presented and discussed.

## 2. Materials and Methods

### 2.1. Starting Materials

*β*-Eucryptite solid solution powders were synthesized following the route proposed (mixture of the precursors) in a previous work (see [6] for details). The chemical compositions of the LAS powder correspond to Li_2_O:Al_2_O_3_:SiO_2_ (1:1.1:2.5) (composition LAS8 in [6]).

The graphene oxide used was provided by GRAPHENEA S.A, (purity 99.99%). This graphene oxide was synthesized by a modified Hummers method [Gr1, Gr2, Gr3].

### 2.2. Powder Mixture Preparation

In order to prepare uniform and well-dispersed nanocomposites, a colloidal processing route was used. The graphene oxide (GO) was supplied as slurry (3.2 wt % in distilled water).

10 g of *β*-eucryptite powder were added into 100 mL of water and it was dispersed under continuous stirring for 30 min. Water suspensions of GO with two different contents (0.5 and 1 wt %) were gradually added into the 10 g of *β*-eucryptite suspensions under continuous magnetic stirring and then sonicated for 30 min in an ultrasonic probe (UP 400 S, Hielscher, Germany). The final suspension was frozen in a liquid nitrogen bath and then dried in a freeze drier (Cryodos-50, Telstar, Spain) for 24 h. The dried powder mixture was ground and sieved using 60 mesh.

### 2.3. LAS-GO Powders Microwave Sintering

A single mode cylindrical cavity operating in the TE_111_ mode with a resonant frequency of 2.45 GHz was selected as the heating cell for microwave. The electric field vectors are perpendicular to the cavity axis with the maximum electric field magnitude at the center, where the samples are located. LAS-GO powders with 0.5 and 1.0 wt % of GO were sintered under vacuum using this microwave technology (see [8,28]). Cylindrical specimens of 10 mm diameter and about 5 mm height were prepared by cold isostatic pressing (200 MPa). These specimens have been located into a quartz tube above the thermal insulator (fiber of alumina). The quartz tube has a 27 mm of diameter and 100 mm of height. Both quartz and alumina are transparent to microwave radiation. These samples were sintered at 1200 and 1250 °C using the heating rate of 100 °C/min with 10 min of holding time at the maximum temperature. The temperature of the sample was monitored by an infrared radiation thermometer (Optris CT-Laser LT, 8−14 μm), which was focused on the test sample via the small circular aperture in the wall of the test cell. In order to compare the composites with monolithic material, LAS samples were sintered using the same microwave conditions.

### 2.4. Characterization Methods

#### 2.4.1. Carbon Content Measurement

The carbon, sulphur, hydrogen and nitrogen content were determined by combustion of 1 mg LAS-GO powders at 1050 °C (LECO CHNS-932, LECO, St Joseph, MI, USA). Under these conditions carbon is transformed into CO_2_, H_2_O and hydrogen sulphur as SO_2_. The oxygen content was determined using a graphite furnace coupled with LECO VTF 900 equip. In total, 1 mg of LAS-GO powders composite was pyrolyzed to 1350 °C under a helium flow of 225 mL·min^−1^. The generated CO oxidizes to CO_2_ forming CuO, which was evaluated and thereby the oxygen content was obtained directly.

The composites sintered at different temperatures were found to have the following graphene contents: 0.45 and 0.90 wt % to LAS + 0.5 wt % GO and LAS + 1.0 wt % GO, respectively.

#### 2.4.2. Raman Characterization

Raman spectra were recorded using a Confocal Raman Microscope (WITec, Ulm, Germany) with a 532 nm excitation laser. Up to 20 spectra were recorded along the whole thickness of the polished composites. The sintered samples were previously cut longitudinally in half cylinders with a diamond saw and polished (Struers, model RotoPol-31) to 0.25 μm using SiC paper and diamond suspension.

#### 2.4.3. Density Measurements

The bulk density of the samples was measured by the Archimedes method (ASTM C373-88) with ethanol as the immersion medium using densities of 2.39 and 2.01 g·cm^−3^ for LAS and graphene, respectively. The relative density was calculated by dividing the bulk density with the theoretical density of the powder mixture.

#### 2.4.4. Mechanical Characterization

Nanomechanical properties such as hardness and Young’s modulus of the sintered composites were obtained by the nanoindentation technique (Model G200, MTS Company, Eden Prairie, MN, USA). To carry out indentations at very low depths, a Berkovich diamond tip was used with radius less than 20 nm. In order to ensure the quality of the tip throughout the experiment, pre- and post- calibration procedures were performed for this indenter ensuring the correct calibration of its function area and correct machine compliance. The nanomechanical properties of the sintered samples were evaluated from the load-displacement nanoindentation data using the widely accepted Oliver and Pharr model [29].

#### 2.4.5. Morphology, Thermal and Electrical Characterization

The fracture surface sections and microstructure of the sintered samples have been studied using a Field Emission Gun Scanning Electron Microscope (FE-SEM, GEMINI ULTRA 55 MODEL, ZEISS, Universitàt Politècnica de València). Specimen microstructures were thermal etching during 30 min at 100 °C below the maximum temperature to reveal grain boundaries. The average grain sizes were measured using the linear interception from FE-SEM micrographs. Approximately 100 grains for each phase were studied [30]. The coefficient of thermal expansion, CTE, was checked in a Netzsch DIL−402-C (Netzsch, Selb, Germany) between −150 and 400 °C. Electrical properties of the sintered samples were measured with a four-point probe.

## 3. Results

A complete Raman spectroscopy study was performed to evaluate and optimise the graphene thermal reduction by microwave radiation. Raman spectroscopy is a very useful technique in order to evaluate the thermal reduction of the graphene oxide (labelled as rG) along the composite.

Electrical resistivity values of LAS and LAS-rG materials are summarized in Table 1. As a result of the reduction process, non-conductive graphene oxide was transformed into a conductive material.

Table 2 shows the relative densities of the LAS monolithic material and the LAS-reduced graphene oxide (LAS-rG) composites sintered at different temperatures by microwave technique.

Density values were very high and close to the theoretical values for monolithic LAS materials [7]. However, the relative densities measured for the composites show a decrease as the graphene content increases.

Some mechanical tests were performed in order to study the behaviour of the composites obtained by microwave processing. The nanoindentation technique shows a common trend clearly detected in the behaviour of hardness (*H*) and Young’s modulus (*E*) values as a function of penetration depth.

To evaluate the dispersion of graphene inside the ceramic matrix, the samples have been examined by electron microscopy. Figure 1 shows the FE-SEM images of the fracture surfaces of the samples sintered by microwaves. *β*-eucryptite grains are well defined in the LAS materials (Figure 1a,b). In the composite images (Figure 1c–f), *β*-eucryptite grains and graphene sheets are observed.

The coefficient of thermal expansion (CTE referred to 25 °C) of the samples sintered by microwave at 1200 °C and 1250 °C is presented in Table 3. The temperature range for these measurements is from cryogenic temperatures at −150 °C to 450 °C. The interest of including cryogenic temperatures in this measurement lies in the spatial applications of *β*-eucryptite materials such as potential substrates in space satellite mirrors.

## 4. Discussion

Figure 2 shows the Raman spectrum that corresponds to the LAS + 0.5 wt % rG and LAS + 1 wt % rG samples sintered by microwave radiation at 1200 °C. The two main Raman features arising from the first-order scattering of the E_2g_ phonon of sp^2^ C atoms (G peak) and the breathing mode of k-point photons of A_1g_ symmetry (D peak), respectively, were observed in all the samples [31]. The effects of the reduction of graphene oxide during the sintering approaches on the Raman characteristics show that both D and G bands undergo significant changes and confirm the successful reduction of graphene oxide by the sintering approaches of LAS-rG composites. Additionally, two small peaks were observed for all composites: a 2D band sensitive to the aromatic C-structure at around 2695 cm^−1^ and an additional peak at 2945 cm^−1^ ascribed to the G + D combination mode induced by disorder or the D + D’ band [32]. The results indicate that the thermal reduction of the graphene oxide during the microwave sintering process at 1200 °C is favored.

As a result of the reduction process, non-conductive graphene oxide was transformed into a conductive material. Electrical resistivity values of LAS and LAS-rG materials are summarized in Table 1. The composites showed isotropic electrical resistivity behaviour due to the fact that the microwave process does not apply any directional pressure during the cycle of sintering as is the case in other processes, e.g., spark plasma sintering (SPS) [19].

The addition of very small amounts of graphene into the LAS matrix leads to an exponential increase in the electrical conductivity of the composite in comparison to the monolithic LAS.

The percolation threshold of the as prepared composites was found to be around 1 wt %, therefore, when the graphene content increases there is an increase in the intersheet connections. In the case of CNTs or CNFs, quantities around 7–10 vol% are needed in order to make the ceramic composites electrically conductive [22]. The connection between CNTs or CNFs is of point to point contact type that leads typically to a high resistance whereas as graphene is a 2D material the connection is of area to area contact type that results in an increased probability of contact between flakes and, as a consequence, a lower electrical resistivity.

The value of electrical resistivity obtained (~4 Ω·cm) with LAS + 1 wt % rG sintered by microwave (in both temperatures), is good enough to shape the nanocomposite using the electro discharge machining (EDM) technique [33,34]. This technique can be an effective alternative for manufacturing complex shape components from hard materials however a minimum electrical conductivity value is required (<100 Ω·cm) [35].

Carbon materials are, in general, very good absorbents of microwaves, i.e., they are easily heated by microwave radiation. This characteristic allows them to be transformed by microwave heating to give rise to new carbon materials or to be used as microwave receptors, in order to heat other materials indirectly, or to act as catalysts and microwave receptors in different heterogeneous reactions. As a consequence, in recent years, the number of processes that combine the use of carbon materials and microwave heating has increased in comparison to other methods that are based on conventional heating [17,18].

The blank hypothesis to be tested in this work is that graphene could have a great influence on the microwave sintering behavior of LAS composites, and therefore, this may mean an increase in their final density, even at a lower temperature. In previous works, it was found that the LAS is an excellent material to absorb the microwave energy and, on the other hand it is known that the graphene has a high dielectric loss factor compared with the LAS ceramic [11,36].

From the results obtained, it is clear that graphene as a second phase directly affects the resulting densities of the studied LAS composites. This can be due to some important factors: the poor adherence of graphene with the LAS matrix, the lack of heat transfer by conduction from the graphene to the ceramic matrix, the agglomeration of graphene or the absence of the microplasma effect [18]. We will now analyze these hypotheses.

The microplasma effect would cause an increase in the local temperature of the graphene platelets. Thus, graphene platelets should be heated rapidly due to (a) high absorption of microwave radiation and (b) transfer of heat to the LAS matrix. Due to the nature of volumetric heating of microwaves, all the graphene platelets might simultaneously absorb microwaves and the temperature of the LAS particles that surround the graphene should also increase. Hence, the heating rate of the mixture would be far better than the one of the pure components, and the positive result (and expected) will be an increase of the density values even at lower temperatures than the one of the dense LAS material (1200 °C). This evidently has not happened. Therefore, the graphene is not acting as a susceptor during the sintering process, as initially thought.

To verify this fact, the sintering of the same material using conventional heating at 1200 °C in vacuum with 2 h of holding time has been carried out. The obtained density values were: LAS (89.5%), LAS + 0.5 wt % rG (90.2%) and LAS + 1.0 wt % rG (91.3%). The density values were very low for all the materials, even the LAS composites with graphene that showed similar values to the ones obtained by microwave processing. Therefore, it has been shown that despite having a high dielectric loss factor and be an excellent electrical conductor, graphene does not act as a good susceptor and does not improve the sinterability of the matrix. An important conclusion is that there was no effect due to the microplasma generated by the graphene in a microwave environment.

The next figure (Figure 3) draws the evolution of the hardness (*H*) and Young’s modulus (*E*) with penetration depth. The large dispersion of *H* and *E* for the initial 200 nm of penetration depth could be due to the implicit experimental variability of factors such as tip-sample interactions, sample roughness and tip rounding. As expected, the low densities recorded for the different composites, are reflected in a sharp drop in the mechanical properties compared to those achieved in monolithic LAS materials. At 1200 °C, the composites with 0.5 wt % rG, the mechanical values tend towards the same value with the depth test. However, composites with 1 wt % rG, both *E* and *H* show a greater dependence with the penetration depth and sintering temperature. Increasing the sintering temperature up to 1250 °C, results in a slight improvement of the mechanical properties, closer to the monolithic LAS. In both cases and keeping in mind the density values, is possible to affirm that the graphene as a second phase in composites leads to a decrease in the mechanical properties, proportional with the quantity of graphene. These data also confirm that graphene is not acting as a susceptor during microwave heating.

Figure 1c–f shows the dispersion of the graphene sheets in the LAS matrix is quite homogeneous. It is observed that there is a clear difference in the *β*-eucryptite grain size reached depending on the presence of graphene, and their percentage within the matrix. The grain size of LAS decreases with the graphene content, i.e., graphene inhibits the grain growth of the matrix, and this grain in the composites is less defined in comparison to the monolithic material. The grain size of the starting powder of LAS is, approximately, 1.0–1.5 µm and the *β*-eucryptite grain size reached in the monolithic material by microwave is less than 2 µm [7,8]

Up to now, the graphene that should be the responsible for the microwave absorption due to a high dielectric loss factor and so, generate heat and heat the other constituent(s) by conventional modes of heating, is the one that causes the poor adherence between the graphene sheets and the ceramic matrix leading to poor mechanical properties in the composites. A possible explanation, which should be thoroughly studied, in order to explain the bad results obtained in the properties of the composites sintered by microwave radiation, is the change in the crystal structure of graphene when a perpendicular electric field was applied [37]. Some authors argue that applying a perpendicular electric field breaks the sublattice symmetry differently depending on the stacking configuration, and thus it is capable of re-ordering the energy hierarchy of the stacking configurations [38,39]. As a consequence, multilayer graphene exhibits the rare behaviour of crystal structure modification, and hence modification of electronic properties, through the application of an external electric field. With the application of a large enough electric field, the crystal structure of graphene passes quickly to on-off conduction [37]. The change in the crystal structure of graphene could also modify the dielectric properties of the graphene and change the heating mechanism of the graphene as a second phase in the ceramic matrix.

This is a supposition or proposed explanation made on the basis of limited evidence as a starting point for further investigation.

The Table 3 shows the CTE values that are negative and very close to zero, in all cases. It is observed that as the content of graphene in the composites increases, the CTE value slightly approaches zero.

Su et al. [40], measured values of CTE of graphene sheets obtained by drying graphene suspensions in a watch glass. CTE values measured in a temperature range of 30–300 °C, showing a negative performance over the entire range with two distinct trends: 30 °C to 160 °C a constant negative value of −6.7 × 10^7^ K^−1^ is obtained and 160 °C to 300 °C the value drops to −1.208 × 10^9^ K^−1^. Therefore, it would be expected that adding graphene in a matrix could lead to more negative CTE values.

CTE values for composites obtained by microwave technology do not exhibit the expected behavior. In the case of the composites with 0.5 wt % of rG, the CTE values are slightly lower than those obtained for the LAS monolithic material for all the sintering temperatures. In the case of composites with 1 wt % of rG, the CTE values present the same order as those obtained for the LAS material.

## 5. Conclusions

The study of the behavior of graphene as a second phase in LAS matrix composites obtained by a non-conventional microwave sintering technique has been carried out for the first time, to the best of our knowledge, in the current investigation. The graphene oxide has been reduced in a one-step by microwave radiation, and therefore, the LAS-rG composites show a high electrical conductivity with very low graphene contents. The dilatometric data, including cryogenic temperatures, show a controlled CTE with very low thermal expansion behavior at all temperatures. The addition of graphene as a second phase, leads to near-zero values. These characteristics could lead to a variety of ceramic materials to suit engineering applications.

Nevertheless, the possible potential of graphene to act as a susceptor and help to heat others dielectric materials by microwave technique is null. The mechanisms and effects involved in the graphene during the application of a perpendicular electric field are still unknown. From these results it can be concluded that graphene, despite having a high dielectric loss factor and being a good candidate to contribute in the sintering process, does not improve the density and the final mechanical properties such as hardness and Young’s modulus of LAS-rG composites. This work will enrich the current graphene-microwave knowledge.

## Figures and Tables

**Figure 1 materials-13-01119-f001:**
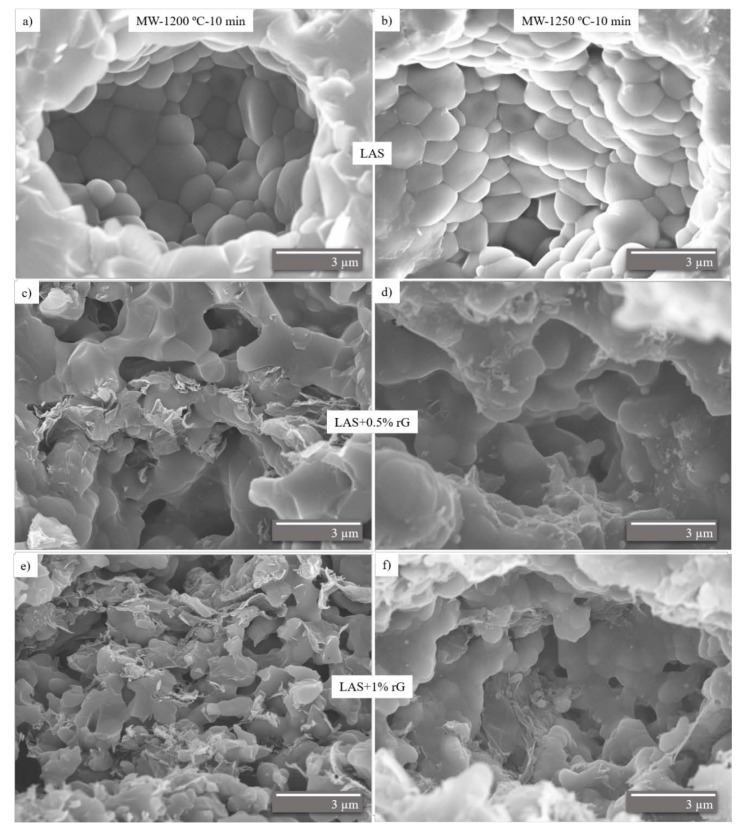
FE-SEM images of the fracture surface of the materials sintered by microwave at 1200 °C (**a**,**c**,**e**) and 1250 °C (**b**,**d**,**f**).

**Figure 2 materials-13-01119-f002:**
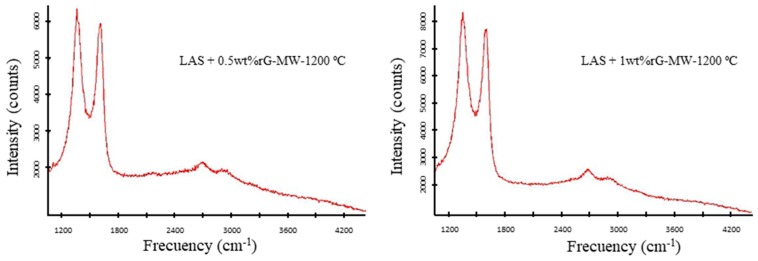
Raman spectrum of the LAS + 0.5 wt % rG and LAS + 1 wt % rG composites after sintering by microwave at 1200 °C.

**Figure 3 materials-13-01119-f003:**
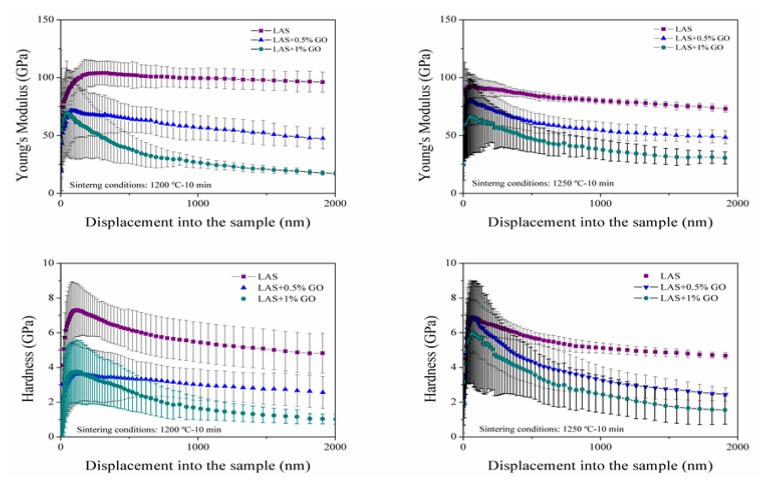
Evolution of the mechanical properties as a function of graphene content and sintering temperature: 1200 °C (**left**) and 1250 °C (**right**).

**Table 1 materials-13-01119-t001:** Electrical characterization of LAS and LAS-rG materials.

Material	Microwave Sintering Temperature (°C)	Electrical Resistivity (Ω·cm)
**LAS**	1200	>10^8^
	1250	>10^8^
**LAS + 0.5 wt % rG**	1200	1.1 ± 0.5·10^4^
	1250	8.7 ± 0.9·10^3^
**LAS + 1 wt % rG**	1200	4.3 ± 0.5
	1250	3.5 ± 0.7

**Table 2 materials-13-01119-t002:** Microwave sintering temperatures and relative density of different materials.

Material	Microwave Sintering Temperature (°C)	Relative Density (%)
**LAS**	1200	99.1 ± 0.4
	1250	99.2 ± 0.4
**LAS + 0.5 wt % rG**	1200	92.5 ± 0.4
	1250	95.1 ± 0.3
**LAS + 1.0 wt % rG**	1200	91.9 ± 0.5
	1250	94.2 ± 0.4

**Table 3 materials-13-01119-t003:** Coefficient of thermal expansion of LAS and LAS-rG materials.

Material	Microwave Sintering Temperature (°C)	CTE −150 +450 °C (10^−6^·K^−1^)
**LAS**	1200	−1.4 ± 0.9
	1250	−1.2 ± 1.2
**LAS + 0.5 wt % rG**	1200	−0.9 ± 0.8
	1250	−0.9 ± 0.9
**LAS + 1 wt % rG**	1200	−0.8 ± 0.7
	1250	−1.4 ± 0.9

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
