# Peer review of "Study of Microwave Heating Effect in the Behaviour of Graphene as Second Phase in Ceramic Composites"

_materials, 2020, doi:10.3390/ma13051119_

Round 1

Reviewer 1 Report

The paper presents modern research, but the article requires many corrections and additions (including repetition of the technological process in optimal conditions - if necessary). Detailed comments there are below.

In what specific applications this group of materials can be used. Huge porosity and numerous pores visible in SEM images significantly limit the possibility of using this type of ceramic material in practice (sometimes it even excludes). First, the entire technological process of the ceramic samples should be optimized. Optimal technological conditions are the most important factor in obtaining materials with desired and assumed properties. The authors write widely about dielectric loss as an important parameter of ceramic materials. However, the authors do not present dielectric tests of the examined group of materials (including dielectric permittivity), which is necessary. line 49 - the authors also write about the need to increase the mechanical strength of ceramic samples (which is to ensure the introduction of graphene). The high porosity of the obtained compositions in the presented SEM images, enhancement of the mechanical strength does not guarantee, point-blank on the contrary. Comparative temperature tests of the electrical conductivity of the resulting compositions would be desirable, and not just their limited measurement (probably at room temperature). line 15 - is “ceramic” should be “ceramics”. line 52 - “…[12].Accordingly, …” missing space. line 90 - the authors define “E” as the electric field vector ., but in line 244 “E” is define like Young’s modulus. Moreover, each constant cited for the first time must be defined (for example line 156). line 97 - there is “8-14 m)” is this correct? line 261 - “Fig. 3c, d, e and f, shows the dispersion of the graphene sheets in the LAS matrix is quite homogeneous.” - but there is no such figure in the manuscript! Did the authors present their hypotheses based on single results or on a larger series of samples? If so, how numerous and were the results and observations reproducible? How repeatable is the method used to obtain samples in order to obtain materials with similar properties? Throughout the text (in most cases) the authors use the degree symbol with an underline. Underlining is not needed. Fig.1 - illegible scale, no markings a, b, c, d, e, f. line 306 - is “ceramics” should be “ceramic”. Fig.2 - no frames are needed, „1200°C” missing space. The figure presents only tests for selected samples, results should be summarized for all. Fig.3 – too small descriptions in figures.

Author Response

(Reviewer 1) In what specific applications this group of materials can be used. Huge porosity and numerous pores visible in SEM images significantly limit the possibility of using this type of ceramic material in practice (sometimes it even excludes). First, the entire technological process of the ceramic samples should be optimized. Optimal technological conditions are the most important factor in obtaining materials with desired and assumed properties.

(Author's Reply) Thanks for this recommendation. The authors will consider this suggestion.

(R1) The authors write widely about dielectric loss as an important parameter of ceramic materials. However, the authors do not present dielectric tests of the examined group of materials (including dielectric permittivity), which is necessary. line 49 - the authors also write about the need to increase the mechanical strength of ceramic samples (which is to ensure the introduction of graphene). The high porosity of the obtained compositions in the presented SEM images, enhancement of the mechanical strength does not guarantee, point-blank on the contrary.

(AR) The authors have already made a study of the dielectric properties of this material and the results have been published, reference [11] in the manuscript: R. Benavente et al. Ceramics International 41(2015) 13817–13822. Lines 219-221.

The carbon material, as graphene, also is an excellent absorber of microwave radiation, this study is referenced in [17,18]. On the other hand, (line 49) one of the hypotheses of this work is to investigation of the graphene as a reinforcement in ceramic materials, as referenced in literature [19,20]. The explication of the lack of improvement of mechanical properties in composites reinforced with graphene is a goal of this study.

(R1) Comparative temperature tests of the electrical conductivity of the resulting compositions would be desirable, and not just their limited measurement (probably at room temperature).

(AR) Yes, it’s interesting, but in this case (composites ceramic-graphene) with the temperature, the second phase (graphene) it deteriorates and the electrical conductivity property with temperature not it’s not real, it would be a measure that would lead to error.

(R1) Line 15 - is “ceramic” should be “ceramics”. line 52 - “… [12]. Accordingly, …” missing space. line 90 - the authors define “E” as the electric field vector., but in line 244 “E” is define like Young’s modulus.

(AR) Thanks for this recommendation. This has been changed in the manuscript.

(R1) Moreover, each constant cited for the first time must be defined (for example line 156). line 97 - there is “8-14 m)” is this correct? line 261 - “Fig. 3c, d, e and f, shows the dispersion of the graphene sheets in the LAS matrix is quite homogeneous.” - but there is no such figure in the manuscript! Did the authors present their hypotheses based on single results or on a larger series of samples?

(AR) The authors evaluated the dispersion of graphene inside the ceramic matrix throught study with several FESEM images. This task not is easy. The images are carefully analyzed and measured. Visually a thorough study is done.

(R1) If so, how numerous and were the results and observations reproducible?

(AR) The observations were, in general terms, quite reproducible for that we can to explain the results.

(R1) How repeatable is the method used to obtain samples in order to obtain materials with similar properties?

(AR) For obtain good and reproducibility results five samples were sintered at each temperature, once the procedure was optimized. For these samples we measure the properties studied and we conclude the results of the manuscript. Therefore, the method used to obtain the samples is repeatable.

(R1) Throughout the text (in most cases) the authors use the degree symbol with an underline. Underlining is not needed.

(AR) Totally agree. It is the typeface of the journal.

(R1) Fig.1 - illegible scale, no markings a, b, c, d, e, f. line 306 - is “ceramics” should be “ceramic”. Fig.2 - no frames are needed, „1200°C” missing space. The figure presents only tests for selected samples, results should be summarized for all. Fig.3 – too small descriptions in figures.

(AR) Thanks for this recommendation. This has been changed in the manuscript. In the Figure 2, the data of Raman spectrum for samples sintered at 1250 ºC is the same of samples sintered at 1200 ºC, hence, the authors have omitted this spectrum, for not repeat the information. The samples sintered at 1250 ºC have successful reduction of graphene oxide.

Reviewer 2 Report

This work is original and very interesting for the microwave sintering community.

However, I have some recommendations listed below:

Line 89: A scheme of the cavity could help to understand this study.

Line 92: Why using vacuum and not a gas like argon, for example? Is there any plasma occurring with this configuration?

Line 98: What is the configuration of the test cell? What thermal insulation is used?

Line 161: SEM images should be more used and detailed. Authors should explained more deeply how they measured the dispersion.

Line 175, Table 3: Is there any comparison possible with data after conventional sintering?

Line 267: How was the grain size measured?

Author Response

This work is original and very interesting for the microwave sintering community.

However, I have some recommendations listed below:

(Review 2) Line 89: A scheme of the cavity could help to understand this study.

(Author's Reply) Thanks for this recommendation. In the text has been cited two references where are the scheme of this cavity and their explication. In the references 8 and 28, there is all information.

(R2) Line 92: Why using vacuum and not a gas like argon, for example? Is there any plasma occurring with this configuration?

(AR) Yes, in this configuration occurring a plasma and is impossible to sinter the materials.

(R2) Line 98: What is the configuration of the test cell? What thermal insulation is used?

(AR) The configuration of the test cell is a tube of quartz of 27 mm of diameter and 100 mm of height. And the specimens have a 10 mm diameter and about 5 mm height.

The thermal insulation is a fiber of alumina. Alumina is transparent to microwave and the porous fiber allow that the sample can heating at high temperatures (>1800 ºC), without affect to tube of quartz.

(R2) Line 161: SEM images should be more used and detailed. Authors should explain more deeply how they measured the dispersion.

(AR) The authors evaluated the dispersion of graphene inside the ceramic matrix thought study a several FESEM images. This task not is easy. The images are carefully analyzed and measured. Visually a thorough study is done.

(R2) Line 175, Table 3: Is there any comparison possible with data after conventional sintering?

(AR) In this reference of the authors, is possible to compared the LAS materials with conventional sintering.

  1. Benavente, M. D. Salvador, O. García-Moreno, F. L. Peñaranda-Foix, J. M. Catalá-Civera, A. Borrell, Microwave, spark plasma and conventional sintering to obtain controlled thermal expansion b-eucryptite materials, International Journal of Applied Ceramic Technology, vol. 12 [S2], E187-E193, 2015.

(R2) Line 267: How was the grain size measured?

(AR) Specimen microstructures were characterized by means of field emission-scanning electron microscopy. A 30 min thermal etching was performed at 100 °C below the maximum temperature to reveal grain boundaries. The average grain sizes were measured using the linear interception from FE-SEM micrographs (ASTM, 2013). Approximately 100 grains for each phase were studied.

(ASTM, ASTM E112-13: standard test methods for determining average grain size, ASTM Int (2013) 1–28, https://doi.org/10.1520/E0112-13.1.4.)

Round 2

Reviewer 1 Report

After the authors' replies to the review, I think that the manuscript may be published in the Materials. However, I found a few minor errors that need improvement.

  • line 262 - “Fig. 3c, d, e and f, shows the dispersion of the graphene sheets in the LAS matrix is quite homogeneous.” – according to the authors' explanations, it is about Figure 1, not applicable to Figure 3.
  • Throughout the text the authors use the degree symbol with an underline. Underlining is not needed. I don't know, maybe the editorial board accepts this entry.
  • Fig.1 has not been corrected and still shows irregularities: illegible scale (maybe using a black background would be appropriate), no markings a, b, c, d, e, f inside figures. It’s important in connection with further description in the text.
  • Fig.2 - too small descriptions in figures.
  • All symbols and constants (for examples: H, E, β) through the text should be written in italics.

Author Response

Study of microwave heating effect in the behaviour of graphene as second phase in ceramic composites

Review 1 Round 2

Comments and Suggestions for Authors

(Reviewer 1) After the authors' replies to the review, I think that the manuscript may be published in the Materials. However, I found a few minor errors that need improvement.

line 262 - “Fig. 3c, d, e and f, shows the dispersion of the graphene sheets in the LAS matrix is quite homogeneous.” – according to the authors' explanations, it is about Figure 1, not applicable to Figure 3.

(Author's Reply) Of course, it was a mistake not to change it in the review. It is already modified. Thanks again.

(R1) Throughout the text the authors use the degree symbol with an underline. Underlining is not needed. I don't know, maybe the editorial board accepts this entry.

(AR) We have changed it, hopefully the publisher accepts the change.

(R1) Fig.1 has not been corrected and still shows irregularities: illegible scale (maybe using a black background would be appropriate), no markings a, b, c, d, e, f inside figures. It’s important in connection with further description in the text.

(AR) We are sorry, in the previous review, we changed the figure but we did not include it in the text, only in the complementary files. It is already modified.

Fig.2 - too small descriptions in figures.

(AR) Thanks for this recommendation. They are already modified.

(R1) All symbols and constants (for examples: H, E, β) through the text should be written in italics.

(AR) Thanks for this recommendation. They are already modified.
